# Serpin-4 Negatively Regulates Prophenoloxidase Activation and Antimicrobial Peptide Synthesis in the Silkworm, *Bombyx mori*

**DOI:** 10.3390/ijms25010313

**Published:** 2023-12-25

**Authors:** Xingtao Qie, Xizhong Yan, Wentao Wang, Yaya Liu, Lijun Zhang, Chi Hao, Zhiqiang Lu, Li Ma

**Affiliations:** 1Department of Plant Protection, College of Plant Protection, Shanxi Agricultural University, Jinzhong 030801, China; qxtjy521@sxau.edu.cn (X.Q.); yanxizhong80@163.com (X.Y.); wwt_200125@163.com (W.W.); liuyaya6112002@163.com (Y.L.); butterflycoco@163.com (L.Z.); sxauhc@163.com (C.H.); 2Department of Entomology, College of Plant Protection, Northwest A&F University, Xianyang 712100, China

**Keywords:** serpin, immune system, prophenoloxidase, antimicrobial peptide, *Bombyx mori*

## Abstract

The prophenoloxidase (PPO) activation and Toll antimicrobial peptide synthesis pathways are two critical immune responses in the insect immune system. The activation of these pathways is mediated by the cascade of serine proteases, which is negatively regulated by serpins. In this study, we identified a typical serpin, BmSerpin-4, in silkworms, whose expression was dramatically up-regulated in the fat body and hemocytes after bacterial infections. The pre-injection of recombinant BmSerpin-4 remarkably decreased the antibacterial activity of the hemolymph and the expression of the antimicrobial peptides (AMPs) *gloverin-3*, *cecropin-D*, *cecropin-E*, and *moricin* in the fat body under *Micrococcus luteus* and *Yersinia pseudotuberculosis serotype* O: 3 (*YP III*) infection. Meanwhile, the inhibition of systemic melanization, PO activity, and PPO activation by BmSerpin-4 was also observed. Hemolymph proteinase 1 (HP1), serine protease 2 (SP2), HP6, and SP21 were predicted as the candidate target serine proteases for BmSerpin-4 through the analysis of residues adjacent to the scissile bond and comparisons of orthologous genes in *Manduca sexta*. This suggests that HP1, SP2, HP6, and SP21 might be essential in the activation of the serine protease cascade in both the Toll and PPO pathways in silkworms. Our study provided a comprehensive characterization of BmSerpin-4 and clues for the further dissection of silkworm PPO and Toll activation signaling.

## 1. Introduction

Lacking adaptive immune responses, insects rely on an innate immune system and physiological barriers to defend themselves against pathogens and parasites [1,2]. The innate immune system is comprised of humoral and cellular immune responses, which function synergistically [1]. The production of antimicrobial peptides (AMPs) is governed by the Toll and IMD pathways, which are two NF-κB signal transduction pathways [1,2]. AMPs are mainly synthesized in the fat body and then secreted into the hemolymph to control systemic infection [3,4,5]. Melanization, which is catalyzed by active phenoloxidase (PO), is a universal defense response to invading pathogens in arthropods [1,6,7,8]. Melanization and the synthesis of AMPs are two key responses in insect humoral immunity. Normally, PO exists in the prophenoloxidase (PPO) zymogen form, which is inactive and needs to be activated through proteolytic cleavage [9,10]. The activation of the PPO and Toll pathways is mediated by the cascade of serine proteases, which is negatively regulated by serine protease inhibitors (serpins) [8,10,11,12,13,14,15].

Serpins, which have a highly variable primary structure but similar conserved three-dimensional structures, can be found in the hemolymph of arthropods with serine proteinase inhibitory activity [11,16,17]. Most serpins are folded into three β-sheets (A-C) and eight or nine α-helical linkers with 350–450 amino acids [16]. β-sheets A and C are connected by an exposed reactive center loop (RCL) of about 20 amino acids, which interacts with the active site of a serine proteinase, leading to a stable serpin–proteinase complex [11,16,18]. Thus, serpin’s inactivation of a proteinase is a suicide mechanism, and the RCL acts as bait for the target protease. In detail, after the RCL interacts with the active sites of a serine proteinase, serpin is cleaved at a scissile bond (P1-P1′) located in the RCL, and most of the serpin adopts a relaxed conformation, which distorts the active site and traps it in the covalent serpin–proteinase complex [16,18]. Serpin genes have been identified systematically in *Drosophila melanogaster* [19], *Manduca sexta* [20], *Bombyx mori* [21], and *Anopheles gambia* [22,23]. Systematic functional studies have been conducted on a series of serpins in *D. melanogaster* through genetic means and in *M. sexta* using biochemical methods.

*Bombyx mori* is a very economically important insect for the production of silk and is a model for the study of genetics, molecular biology, and immunology in Lepidoptera insects [24]. Thirty-four serpin genes have been identified in silkworm, and their molecular characterization and expression patterns in response to microbial infection have been profiled [21]. Functional studies for some serpins have been reported. Bmserpin-15 was up-regulated significantly after being challenged with different microorganisms, particularly fungus and Gram-positive bacteria, and Bmserpin-15 negatively regulated the activation of the PPO pathway and the synthesis of AMPs through the Toll pathway [25]. Bmserpin-16 was specifically expressed at high levels in the silk glands, and it showed inhibitory activity against cysteine proteinase, while the physiological function of Bmserpin-16 in the silk glands is unclear [26]. The obvious up-regulation of Bmserpin-28 was observed following pathogen challenges in the fat body and hemocytes, and the knockdown of *Bmserpin-28* expression through RNA interference resulted in the significant up-regulation of AMP genes [27]. Our previous study revealed that Bmserpin-5 inhibited the serine protease cascade during the activation of the PPO and Toll pathways through the targeting of BmHP6 and BmSP21, leading to the obstruction of the encapsulation and melanization of beads, systemic melanization, and the synthesis of AMPs after bacterial challenges [17]. 

Bioinformatics analysis showed that BmSerpin-4 is a typical serpin protein that might play a role in the immune system of *B. mori* [21]. In this study, the biochemical function of BmSerpin-4 in the immune responses of *B. mori* was revealed. It was shown that the expression of *BmSerpin-4* in the fat body and hemocytes was induced by the immune challenge from Gram-negative and Gram-positive bacteria. The significant inhibition of systemic melanization, PO activity, and the activation/clipping of PPO by BmSerpin-4 was revealed. Meanwhile, remarkable inhibition of antibacterial activity by BmSerpin-4 was detected in the hemolymph. Substantial decreases in induced AMPs, gloverins, cecopins, and moricin, were identified in the bacterially challenged hemolymph due to the effect of BmSerpin-4. Quantitative real-time PCR further confirmed that *gloverin-3*, *cecropin-D*, *cecropin-E*, and *moricin* were negatively regulated by BmSerpin-4 after bacterial challenge. Hemolymph proteinase1 (HP1), serine protease 2 (SP2), HP6, and SP21 were predicted as the candidate target serine proteases for BmSerpin-4 through the analysis residues adjacent to the scissile bond and comparisons of orthologous genes. It was suggested that HP1, SP2, HP6, and SP21 might participate in the serine protease cascade, leading to the activation of the Toll and PPO pathways. This study provided a comprehensive characterization of BmSerpin-4 and clues for the further dissection of silkworm PPO and Toll activation signaling.

## 2. Results

### 2.1. Sequence, Domain Identification, Three-Dimensional Structure, and Phylogenetic Assays of BmSerpin-4

BmSerpin-4 (NP_001037090) was identified as a 410 amino acid (aa) protein, with a predicted signal peptide that consisted of the first 17 aa residues (Figure 1A). The calculated molecular weight and isoelectric point of the mature protein were 44.44 kDa and 6.56, respectively. Functional domain analysis indicated that BmSerpin-4 contained a typical serpin domain (34–402 aa) and its reactive center loop was located in the C terminal (360–379 aa) (Figure 1A). According to the sequence alignment of the RCL regions in *B. mori* serpins [21], the scissile bond (P1-P1′) was predicted to be Arg-Ile (R-I). Furthermore, the obtained three-dimensional structure model was conclusive, revealing that BmSerpin-4 had the typical structure of three (A-C) β-sheets and nine α-helical linkers, and the A and C β-sheets were linked by the RCL (Figure 1B). Thus, we identified BmSerpin-4 as a typical serpin protein. Phylogenetic analysis revealed that BmSerpin-4 and *M. sexta* Serpin-4 (MsSerpin-4) are in the same clade (Figure 1C), and BmSerpin-4 is orthologous to MsSerpin-4, with a 65% shared identity within their amino acid sequences. The high similarity between the two serpins suggests that they might share similar functions.

### 2.2. Bacterial Infection Induces BmSerpin-4 Expression in the Fat Body and Hemocytes of Larval Silkworm

The midgut, fat body, and hemocytes are the locations of the major immunity tissues in insects [1]. The expression of *BmSerpin-4* in the midgut, fat body, and hemocytes was examined through q-PCR, and the results showed that *BmSerpin-4* was mainly expressed in the fat body and hemocytes (Figure 2A). The transcriptional level significantly increased in the hemocytes (Figure 2B) and fat body (Figure 2C) after Gram-positive (*M. lutes*) and Gram-negative (*YP III*) bacterial infections. Thus, our results indicate that *BmSerpin-4* in the hemocytes and fat body of *B. mori* responds to bacterial infections. Furthermore, the protein levels of BmSerpin-4 in the hemolymph after infection by *M. lutes* and *YP III* were examined through Western blotting using the prepared BmSerpin-4 antibody. The protein levels of BmSerpin-4 were increased at 6 and 12 hpi (hours post-infection) after bacterial infection, and especially at 24 hpi (Figure 2D).

### 2.3. Purification of the Recombinant BmSerpin-4

To investigate the physiological function of BmSerpin-4, the mature and biologically active recombinant BmSerpin-4 with a 6× His-tag at the N-terminus was expressed and purified from *E. coli*. The recombinant protein was detected through SDS-PAGE stained with Coomassie blue, with the approximate molecular weight being 44 kDa after induction by IPTG (Figure 3A). The soluble fusion protein was successfully purified from the soluble fraction by nickel affinity chromatography (Figure 3A). Western blotting analysis of the induced recombinant protein using an anti-His antibody (Figure 3B) and anti-BmSerpin-4 antibody (Figure 3C) confirmed the consensus protein had a weight of 44 kDa. The purified recombinant protein was further verified as BmSerpin-4 using mass spectrometry.

### 2.4. BmSerpin-4 Inhibits Systemic Melanization in the Larval Silkworm

The larval silkworms were inert, and the body color of the silkworms was darker twelve hours after being injected with *M. lutes* and *YP III*. The plasma collected from the treated silkworms was a dark color, and many dark nodules in the hemocoel were seen after dissection (Figure 4A,C). On the contrary, if the larvae were injected with recombinant BmSerpin-4 before the infection with bacteria, there were no nodules found in the hemocoel and the collected plasma was yellow (Figure 4B,D). Hence, these observations indicate that BmSerpin-4 inhibited systemic melanization induced by the dead bacterial cells. Melanization is catalyzed by active PO; normally, PO exists in the PPO zymogen form, which is inactive and needs be activated through proteolytic cleavage. This suggests that BmSerpin-4 might inhibit PPO activation. 

### 2.5. BmSerpin-4 Inhibits the Activation of PPOs

Generally, pathogen-associated molecular patterns (PAMPs), such as peptidoglycans (PGs) and lipopolysaccharides (LPSs) from bacteria as well as curdlans from fungi, are recognized by pattern recognition proteins (PRPs), and then the sequential activation of the serine protease cascade in the hemolymph of an insect is initiated, resulting in the activation of PPO and melanization [28]. We suspected that the activation of PPO in the silkworm might be negatively regulated by BmSerpin-4 according to the above results. To test this hypothesis, the silkworms were injected with recombinant BmSerpin-4; then, the plasma without hemocytes was collected, and shortly afterward it was treated with different types of PAMPs to trigger immune responses in vitro to detect the PO activity and activation of PPO. The results showed that BmSerpin-4 obviously blocked the PO activity with or without the activation by PAMPs, and spontaneous melanization was also dramatically inhibited by BmSerpin-4 (Figure 5). Furthermore, the cleavage of PPO1 and PPO2 and the formation of a high-molecular-mass PO complex during the process of activation were inspected through Western blotting. The cleavage of PPOs and the formation of the PO complexes were clearly observed in the Western blotting results, where the plasma was collected from the BSA-pre-injected silkworms and then incubated with curdlan (Figure 6A), LPS (Figure 6B), and PG (Figure 6C,D). However, in the presence of BmSerpin-4, the cleavage of PPOs and the formation of the PO complexes both were partially inhibited or abolished completely during the activation of the PAMPs (Figure 6). Overall, these results demonstrate that BmSerpin-4 effectually down-regulated the PPO pathway.

### 2.6. BmSerpin-4 Inhibits AMP Generation Induced by Bacteria

Remarkable antibacterial activity was detected through an inhibition bacterial zone assay 24 h after *M. luteus* and *YP III* infection in the hemolymph. In contrast, in the presence of BmSerpin-4, the antibacterial activity induced by both *YP III* (Figure 7A) and *M. luteus* (Figure 7B) obviously decreased, especially in the hemolymph of the silkworms challenged by *M. luteus* (Figure 7B). Moreover, the composition of the small-molecule peptides in the bacterially challenged hemolymph was examined with tricine-SDS-PAGE. It showed that the proteins induced by *M. luteus* and *YP III* infection were located at about 16 kDa (band 1) and 4 kDa (band 2), while the pre-injection of BmSerpin-4 inhibited their production (especially that induced by *M. luteus*) conspicuously (Figure 7C). Our previous LC/MS-MS analysis identified band 1 as being gloverins and band 2 as being cecropins and moricin [17]. The q-PCR results further confirmed that the expression of *gloverin-3*, *cecropin-D*, *cecropin-E,* and *moricin* was strongly induced in the fat body by *M. luteus* and *YP III*, and their expression was negatively regulated by BmSerpin-4 (Figure 7D). These results together demonstrated that BmSerpin-4 effectually inhibits AMP generation induced by bacteria, suggesting that BmSerpin-4 can negatively regulate the synthesis of AMPs through the Toll pathway.

## 3. Discussion

Serpins widely exist in plants, invertebrates, vertebrates, and even in viruses [29]. For instance, the hemolymphs of insects and other arthropods contain relatively high concentrations of serpins from several different gene families [11]. Serpins, as inhibitors of serine or cysteine proteases through the combination of tight-binding and trapping inhibitors, may be involved in the regulation of multiple immune and other physiological functions, such as the proteolytic activation of cytokines, antimicrobial peptide synthesis, melanotic encapsulation, the clotting of hemolymphs, and wound healing in insects [11,16]. In the present study, BmSerpin-4 was identified as a typical serpin, and the results indicated that BmSerpin-4 effectually inhibited the synthesis of AMPs and the activation of the PPO pathway/melanization. AMPs play a crucial role in the humoral defense responses of insects, and they are mainly synthesized by two NF-κB signal transduction pathways, the Toll and IMD pathways. The activation of Spätzle, the ligand of the trans-membrane Toll receptor, and PPO depend on an extracellular clip domain, which is regulated by serpins [8,10,12,13,14,15,30,31]. So, BmSerpin-4 might inhibit the cleavage and activation of the downstream proteases involved in the two pathways.

It was confirmed that MsSerpin-4 negatively regulated the PPO activation cascade and Toll pathway in *M. sexta* by inhibiting the targeted serine proteases of HP1, HP5, HP6, and HP21 [12,31,32]. In *B. mori*, the serine proteases of HP1, SP2, HP6, and SP21 were orthologous to HP1, HP5, HP6, and HP21 in *M. sexta*, respectively [17]. The comparisons of the sequences, including MsHP1 vs. BmHP1, MsHP5 vs. BmSP2, MsHP6 vs. BmHP6, and MsHP21 vs. BmSP21, showed that they shared 74%, 52%, 65%, and 56% identities within the amino acid residues (Appendix A). The inhibitory selectivity of serpins was largely determined by the residues adjacent to the scissile bond. The scissile bonds of BmSerpin-4 and MsSerpin-4 were both Arg-Ile. The scissile bonds of the targeted serine proteases of MsSerpin-4 in *M. sexta* and their corresponding orthologous proteases in *B. mori* were the same as and similar to Arg-Ile, respectively (Figure 8A). Therefore, the candidate target serine proteases for BmSerpin-4 were predicted to be HP1, SP2, HP6, and SP21. Furthermore, the predicated cleavage sites of BmSerpin-4 and the serine proteases identified as the components of the cascade leading to Toll and PPO in *B. mori*, *M. sexta*, and *D. melanogaster* were compared. *B. mori* BAEEase and PPAE (PAP3) have similar activation sites (Arg-Ile and Lys-Ile, respectively) to BmSpn-4 (Arg-Ile) (Figure 8A), which suggests that BAEEase and PPAE might be cleaved and activated by the target serine proteases of BmSerpin-4. The BAEEase of *Bombyx* exists as a zymogen and can be activated by upstream serine protease cascade components in the presence of PG or β-1,3-glucan, and its homolog in *Drosophila*, the Spätzle-processing enzyme (SPE), was required for a Toll-dependent antimicrobial response [33]. In *Tenebrio molitor*, the SPE-activating enzyme (SAE, the upstream serine protease of SPE) and SPE were essential for the activation of the Toll pathway [34]. The activation site of *B. mori* BAEEase was similar to that of *T. molitor* SPE, *D. melanogaster* SPE, and *M. sexta* HP8, enabling it to serve as the SPE [33,34,35,36] (Figure 8A). In *M. sexta*, HP21 cleaved proPAP2, proPAP3, and proHP5, and HP5 cleaved proHP6 [33,37,38]. Activated *M. sexta* HP6 cleaved proHP8 and proPAP1 [12]. Thus, HP5, HP6, and HP21 were required to activate the PPO and Toll pathways in *M. sexta*. Based on this information and our results indicating that BmSerpin-4 inhibited the synthesis of AMPs, we hypothesized that BAEEase might play a critical role as the SPE and could be activated by the target serine protease, HP6, of BmSerpin-4. The *B. mori* PPO-activating enzyme (PPAE, PAP3) was identified as the ortholog of *M. sexta* PAP3 [39]. ProHP1 could be activated without being cleaved, and proHP1*, instead of HP1, could cut proHP6 to form HP6, generating HP8 and PAP1 to trigger the PPO and Toll pathways [40]. Previously, we also identified BmPAP2 as the ortholog of *M. sexta* PAP2 [17,41]. Based on this information and our results indicating that BmSerpin-4 inhibited the activation of PPO, we hypothesized that HP1, SP2, HP6, and SP21, as the candidate serine proteases, play key roles in not only the activation of the Toll pathway but also in PPO activation (Figure 8B).

In conclusion, this study revealed that BmSerpin-4 negatively regulated PPO activation and AMP synthesis by predictably targeting HP1, SP2, HP6, and SP21 in the silkworm, which is important knowledge for the further dissection of silkworm PPO and Toll activation signaling.

## 4. Materials and Methods

### 4.1. Rearing of Silkworms and Bacterial Infection

The silkworms, *B. mori*, (Nistari strain) were reared on fresh mulberry leaves at 27 °C in 70% relative humidity and under a photoperiod of 13:11 (light–dark). On day 3, the 5th-instar silkworm larvae were used in all experiments. *Micrococcus luteus* and *Yersinia pseudotuberculosis serotype* O: 3 (*YP III*) (gifted by Dr. Xihui Shen from Northwest A&F University) were cultured at 37 °C in LB medium overnight. The cells were collected through centrifugation. The collected pellet was resuspended in 1 mL of sterilized 0.85% NaCl, and then 40 mL of 70% isopropylalcohol was added and incubated for 1 h at room temperature with rotation to kill the bacterial cells. After three rounds of washing with sterilized 0.85% NaCl, the isopropanol-killed bacteria were resuspended in sterilized 0.85% NaCl until OD 600 nm of 10 was reached. Fifty microliters of the bacterial preparations were injected into the hemocoel of the silkworms.

### 4.2. Protein Sequence, Domain Identification, Three-Dimensional Structure, and Phylogenetic Assays

The nucleotide and amino acid sequences of BmSerpin-4 were retrieved from the NCBI database (https://www.ncbi.nlm.nih.gov/ accessed on 5 November 2022). Domains were predicted by searching InterPro (http://www.ebi.ac.uk/interpro/search/sequence/ accessed on 6 May 2023), and annotations of the amino acid sequences were predicted by searching in the NCBI database. The visualization of the three-dimensional structure of BmSerpin-4 was retrieved from the AlphaFold protein structure database (https://alphafold.com accessed on 2 May 2023). After alignment of the selected serpin protein sequences of *B. mori*, *M. Sexta*, and *D. melanogaster* using ClustalW, a phylogram was constructed using the neighbor-joining method in MEGA 5.10.

### 4.3. RNA Extraction, cDNA Synthesis, and Quantitative Real-Time PCR (q-PCR)

The total RNA of each tissue was extracted using the TriPure Reagent (Roche, Basel, Switzerland), according to the manufacturer’s instructions. The resulting total RNA samples were further purified with Direct-zol™ RNA MiniPrep (Zymo Research, Irvine, CA, USA) to remove genomic DNA contamination. First-strand cDNA was synthesized with the Transcriptor First Strand cDNA Synthesis Kit using 1 µg of the purified total RNA. Q-PCR was performed on a Rotor Q thermocycler (Qiagen, Hilden, Germany) by using 2× Faststart Essential DNA Green Master (Roche, Basel, Switzerland) and the prepared first-strand cDNA as the template. At the end of each q-PCR procedure, a melt curve ranging from 65 to 95 °C was performed to confirm the amplification of the specific PCR product. The housekeeping gene *IF4A* (DQ443290.1) of *B. mori* was used as an internal control to normalize the transcript level of *BmSerpin4* and AMP genes [42]. The results were evaluated using a relative quantitative method (2^−ΔΔCt^). All analyses were performed with three biological replicates. The primers used in the q-PCR are listed in Table 1. The R^2^ values of the standard curves were over 0.980 and the calculated amplification efficiency was 90–110%. This indicates that the q-PCR reactions were performed under optimal conditions.

### 4.4. Expression and Purification of Recombinant BmSerpin-4

For recombinant BmSerpin-4 expression, the DNA fragment encoding mature BmSerpin-4 without its signal peptide was amplified through PCR with specific primers, in which *Xho* I and *Bam* HI sites were introduced (Table 1). The PCR product was separated on a 1% agarose gel and purified using a DNA Gel Extraction Kit (Tiangen, Beijing, China). After *Xho* I and *Bam* HI (Takara, Dalian, China) were digested, the purified DNA was ligated into the pET-28a (+) expression vector (Novagen, Madison, WI, USA) using the T4 ligase (Takara, Dalian, China). The recombinant plasmid was transformed into *Escherichia coli* BL21 (DE3) competent cells (Tiangen, Beijing, China), which was confirmed by sequencing. Then, the expression of BmSerpin-4 in the *E. coli* BL21 (DE3) was induced overnight by 0.5 mM isopropyl-beta-D-thiogalactopyranoside (IPTG) at 20 °C and 150 r/min. The bacterial cells were harvested through centrifugation at 4 °C, and the collected cells were resuspended in 20 mM Tris-HCl (pH 8.3) and homogenized through sonication (Sonics Vibra-Cell, SONICS & MATERIALS, Inc., Newtown, PA, USA, 130 w, 30% power) on ice. After centrifugation at 12,000× *g* for 20 min at 4 °C, BmSerpin-4 was purified from the soluble fraction using a Ni^2+^-NTA column with a stepwise elution of 20 mM Tris-HCl (pH 8.3) containing 25, 50, 75, 100, 150, and 500 mM imidazole. The concentration of purified BmSerpin-4 was determined using the EasyII Protein Quantitative Kit (BCA) (TransGen Biotech, Beijing, China), with bovine serum albumin (BSA) used as the standard. Purified BmSerpin-4 samples were aliquoted into Eppendorf tubes and stored at −80 °C.

### 4.5. The Preparation of BmSerpin-4 Polyclonal Antiserum

The BmSerpin-4 polyclonal antiserum was prepared using ABclonal (Wuhan, China). Briefly, 1 ml of the purified BmSerpin-4 protein (0.6 mg/mL) was injected four times into a New Zealand white rabbit at ten-day intervals for immunization. The antiserum was obtained from the immunized rabbit, with the blood collected from the marginal vein of the ear of the pre-immunized rabbit used as a negative control. The protein levels of BmSerpin-4 in the hemolymph after the bacterial infections were examined through Western blotting (WB), with the prepared BmSerpin-4 antiserum (1:5000 dilution) used as the primary antibody.

### 4.6. Phenoloxidase Activity and Prophenoloxidase Activation Assays

The assays were performed as we described previously [17] with slight modifications. The silkworm larvae were injected with 50 µL of purified BmSerpin-4 (200 µg/mL in 20 mM Tris-HCl, pH 8.3) or 50 µL of BSA (200 µg/mL in 20 mM Tris-HCl, pH 8.3). After 30 min, the hemolymph was collected and centrifuged immediately at 16,000× *g* for 30 s at 4 °C to remove the hemocytes. Then, 2 µL of the plasma sample was added to the wells of a 96-well plate, and 2 µL of the pathogen-associated molecular patterns (PAMPs, 1 mg/mL), either being polysaccharides (LPSs) from *Porphyromonas gingivalis*, curdlans from *Alcaligenes faecalis*, or peptidoglycans (PGs) from *E. coli* or *M. luteus*, and 100 µL of 2 mM dopamine was added and then mixed immediately. The absorption at 490 nm was recorded every 30 s for 20 min on the microplate reader (Spectra Max M5, Molecular Devices, San Jose, CA, USA) at 25 °C. PO activity was detected as the maximum slope, which was defined as an increase in absorbance at 490 nm/min [43]. 

Next, 60 µL of each of the two groups of freshly prepared plasma was added simultaneously to each 1.5 mL Eppendorf tube that contained 10 µL of 1 mg/mL elicitor, and the tubes were incubated at room temperature. At 1 min, 5 min, and 25 min, 20 µL of each sample was taken and mixed with 20 µL of 4× SDS loading buffer, and was then boiled immediately at 100 °C for 5 min. After centrifugation at 5000× *g* for 3 min, the samples were analyzed through 8% SDS-PAGE and Western blotting with a mixture of anti-PPO1 and anti-PPO2 (1:5000 dilution) (gifts from Dr. Michael Strand from the University of Georgia) as the primary antibodies.

### 4.7. Hemolymph Antibacterial Activity Assay

The assay was performed as we described previously [17] with some modifications. On day 3, fifth-instar larvae were anesthetized on ice for 15 min, followed by injection with 10 µg of BSA or 10 µg of purified BmSerpin-4 in 50 µL of Tris-HCl (pH 8.3). After 30 min of incubation in a rearing incubator, the larvae were anesthetized on ice for another 15 min. Then, the two groups of silkworms were immune challenged by isopropanol-killed *M. luteus* and *YP III*. Additionally, 50 µL of 0.85% NaCl was injected into another BSA pre-injection group as the control. The fat bodies from at least 5 silkworm larvae were collected 12 h after the bacterial infection. The expression of AMPs was subsequently analyzed using q-PCR using the specific primers of the AMPs (Table 1). The plasma samples were collected and prepared 24 hpi (hours post-infection) as described previously [35]. The plasma antimicrobial activity assay was performed by measuring zones of growth inhibition of *M. luteus* or *YP III* in a thin layer of agar plates, as described previously [44]. Accordingly, 30 µL of each of the above plasma samples was treated with 10 µL of 4× SDS loading buffer at 100 °C for 5 min, and 10 µL of each of the resulting samples was separated by electrophoresis on a 16% tricine-SDS-PAGE [45] after centrifugation at 6000× *g* for 2 min. 

### 4.8. Statistical Analysis

The gene expression, PO activity, and antibacterial assay data were plotted using GraphPad Prism 9.0 (GraphPad Inc., La Jolla, CA, USA). Multiple group comparisons were analyzed using the one-way ANOVA test or the two-way ANOVA test. Comparisons between the control and treatment groups were analyzed using an unpaired Student’s *t*-test (two-tailed) to identify significant differences.

## Figures and Tables

**Figure 1 ijms-25-00313-f001:**
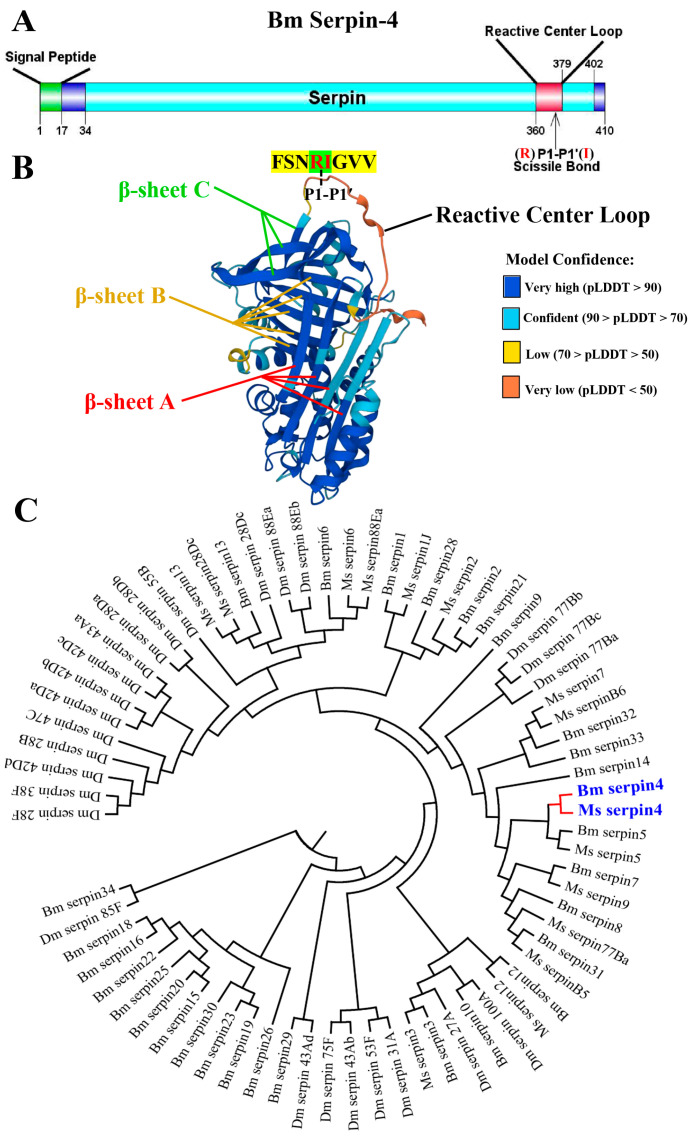
The functional domain (**A**), three-dimensional structure (**B**), and phylogenetic assays of BmSerpin-4 (**C**). (**A**) A predicted signal peptide that consists of the first 17 residues and a serpin domain (34–402 aa) was found in the amino acid sequence of BmSerpin-4 (NP_001037090). A reactive center loop with the predicted P1-P1′ scissile bond (R-I) is located in the C terminal (360–379 aa) of the serpin domain. (**B**) The tertiary structure of mature BmSerpin-4 was retrieved and visualized from AlphaFold protein structure database (https://alphafold.com (accessed on 2 May 2023)). BmSerpin-4 has the typical structure of three (A–C) β-sheets and nine α-helical linkers, and the A and C β-sheets are linked by the reactive center loop. The different colors in the diagram represent the corresponding credibility, as indicated on the right. (**C**) The phylogram was built by a neighbor-joining method using MEGA 5.10 after aligning the selected serpin protein sequences of *B. mori*, *M. Sexta*, and *D. melanogaster* using ClustalW. Bm serpin4: NP_001037090; Ms serpin4: AAS68503.1; Bm serpin5: AAS68506.1; Ms serpin5: AAS68507.1; Bm serpin7: NP_001139701; Ms serpin9: AYK02794.1; Bm serpin8: NP_001139702; Ms serpin77Ba:XP_030032247.2; Bm serpin31: NP_001139722; Ms serpinB5: XP_030040571; Bm serpin12: NP_001036857; Ms serpin12: AYK02795.1; Dm serpin100A: NP_651818.1; Bm serpin10: NP_001139703; Dm serpin27A: NP_001260143.1; Bm serpin3: XP_037867815; Ms serpin3: AAO21505.1; Dm serpin31A: NP_609341.1; Dm serpin35F: NP_001036553.1; Dm serpin43Ab: NP_001027395.1; Dm serpin75F: NP_001036614.1; Dm serpin43Ad: NP_610261.1; Bm serpin29: NP_001139720; Bm serpin26: NP_001139718; Bm serpin19: NP_001139712; Bm serpin23: NP_001139716; Bm serpin30: NP_001139721; Bm serpin15: NP_001139707; Bm serpin20: NP_001139713; Bm serpin25: NP_001139717; Bm serpin22: NP_001139715; Bm serpin16: NP_001139708; Bm serpin18: NP_001139711; Dm serpin85F: NP_649965.2; Bm serpin34: NP_001129364; Dm serpin28F: NP_524957.2; Dm serpin38F: NP_001286112.1; Dm serpin42Dd: NP_001163067.1; Dm serpin28B: NP_001260209.1; Dm serpin47C:NP_001163116.1; Dm serpin42Da: NP_524955.2; Dm serpin42Db: NP_001260746.1; Dm serpin42Dc: NP_001246154.1; Dm serpin43Aa: NP_524805.1; Dm serpin28Da: NP_001036345.1; Dm serpin28Db: NP_001356884.1; Dm serpin55B: NP_524953.1; Ms serpin13: AYK02793.1; Ms serpin28Dc: XP_030027527; Bm serpin13: NP_001139705; Dm serpin28Dc: NP_609172.1; Dm serpin88Ea: NP_524954.2; Dm serpin88Eb: NP_650427.1; Bm serpin6: NP_001103823; Ms serpin6: AAV91026.1; Bm serpin1: NP_001037305; Bm serpin1J:AAC47340; Ms serpin88Ea: XP_037300507; Bm serpin28: NP_001139719; Ms serpin2: AAB58491.1; Bm serpin2: NP_001037021; Bm serpin21: NP_001139714; Bm serpin9: NP_001037530; Dm serpin77Bb: NP_649206.1; Dm serpin77Bc: NP_001262103.1; Dm serpin77Ba: NP_001287128.1; Ms serpin7: ADM86478.1; Ms serpinB6: XP_030039875; Bm serpin32: NP_001139723; Bm serpin33: NP_001129363; Bm serpin14: NP_001139706.

**Figure 2 ijms-25-00313-f002:**
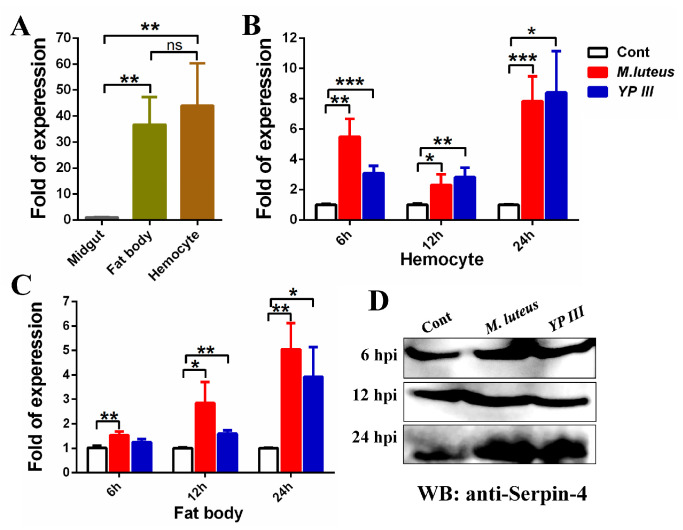
The mRNA abundance of *BmSerpin-4* in the immune tissues (**A**) and the expression of *BmSerpin-4* in hemocytes (**B**) and fat body after bacterial infection (**C**). (**A**) The main immune tissues from the midgut, hemocytes, and fat body were collected to analyze the mRNA abundance of *BmSerpin-4*. Hemocyte (**B**) and fat body (**C**) samples were collected at 6, 12, and 24 hpi (hours post-infection) to analyze the transcription levels of *BmSerpin-4* using q-PCR. The expression of *BmSerpin-4* was normalized with *IF4A* gene of *B. mori*, and the values shown are the means (±SEM) of three independent biological experiments. (**D**) The protein levels of BmSerpin-4 in the hemolymph after infection by *M. lutes* and *YP III* at 6, 12, and 24 hpi were examined through Western blotting (WB). For (**A**), the relative expression of *BmSerpin-4* in different tissues was compared to the expression in midgut, and the statistical differences between compared groups were denoted with asterisks. *p*-values were determined using Student’s *t*-test. ns: no significance; ** *p* < 0.01. For (**B**,**C**), the relative expression of the infection groups was compared to the expression of the control groups at each time point, and the statistical differences between the control groups and infection groups were denoted with asterisks. *p*-values were determined by one-way ANOVA, followed by Tukey’s multiple range test. * *p* < 0.05; ** *p* < 0.01; *** *p* < 0.001.

**Figure 3 ijms-25-00313-f003:**
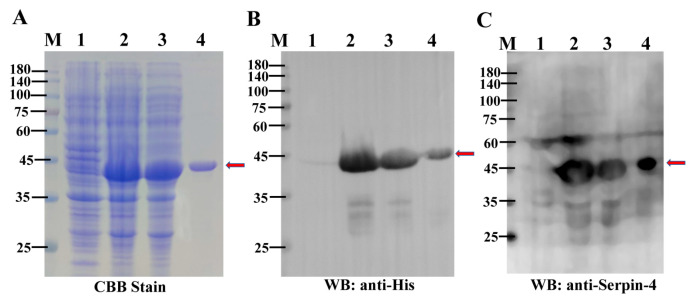
The expression and purification of recombinant Bmserpin-4. (**A**) Coomassie brilliant blue-stained SDS-PAGE. (**B**) Western blot probed with anti-His antibody. (**C**) Western blot probed with anti-BmSerpin-4 antibody. M—protein molecular mass markers; lane 1—total proteins from uninduced *E. coli* cells; lane 2—total proteins from induced *E. coli* cells; lane 3—soluble proteins from induced *E. coli* cells; lane 4—purified recombinant BmSerpin-4. The bands of recombinant Bmserpin-4 are indicated by the red arrows.

**Figure 4 ijms-25-00313-f004:**
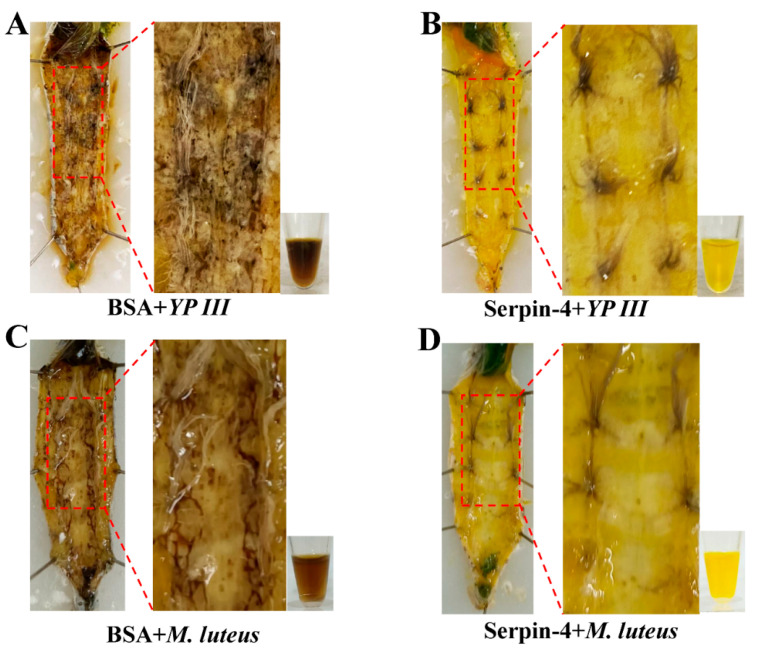
BmSerpin-4 inhibited systemic melanization. The anatomic melanization of tissues and plasma after pre-injection of 10 µg of BSA following approximately 2 × 10^8^ dead *YP III* (**A**) or 1 × 10^8^ dead *M. luteus* infection (**C**). The anatomic melanization of tissues and plasma after pre-injection of 10 µg of BmSerpin-4 following approximately 2 × 10^8^ dead *YP III* (**B**) or 1 × 10^8^ dead *M. luteus* (**D**) infection. The samples were collected and observed 12 h after the injection of bacterial cells.

**Figure 5 ijms-25-00313-f005:**
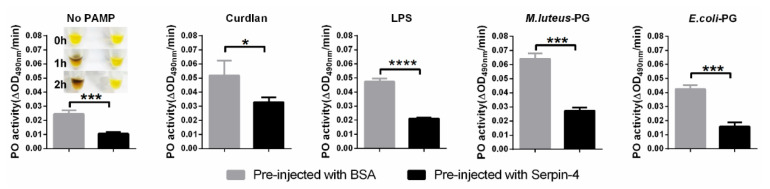
BmSerpin-4 blocked the PO activity and spontaneous melanization of the hemolymph. The spontaneous melanization of the hemolymph collected from the larvae pre-injected with BSA or BmSerpin-4 was observed. The PO activity of the hemolymph from the larvae pre-injected with BSA or BmSerpin-4 and then activated with or without pathogen-associated molecular patterns (PAMPs, including curdlan, LPS, and PG) was measured. The values shown are the means (±SEM) of three independent experiments. The statistical differences between the compared groups are denoted with asterisks. *p*-values were determined using Student’s *t*-test. * *p* < 0.05; *** *p* < 0.001; **** *p* < 0.0001.

**Figure 6 ijms-25-00313-f006:**
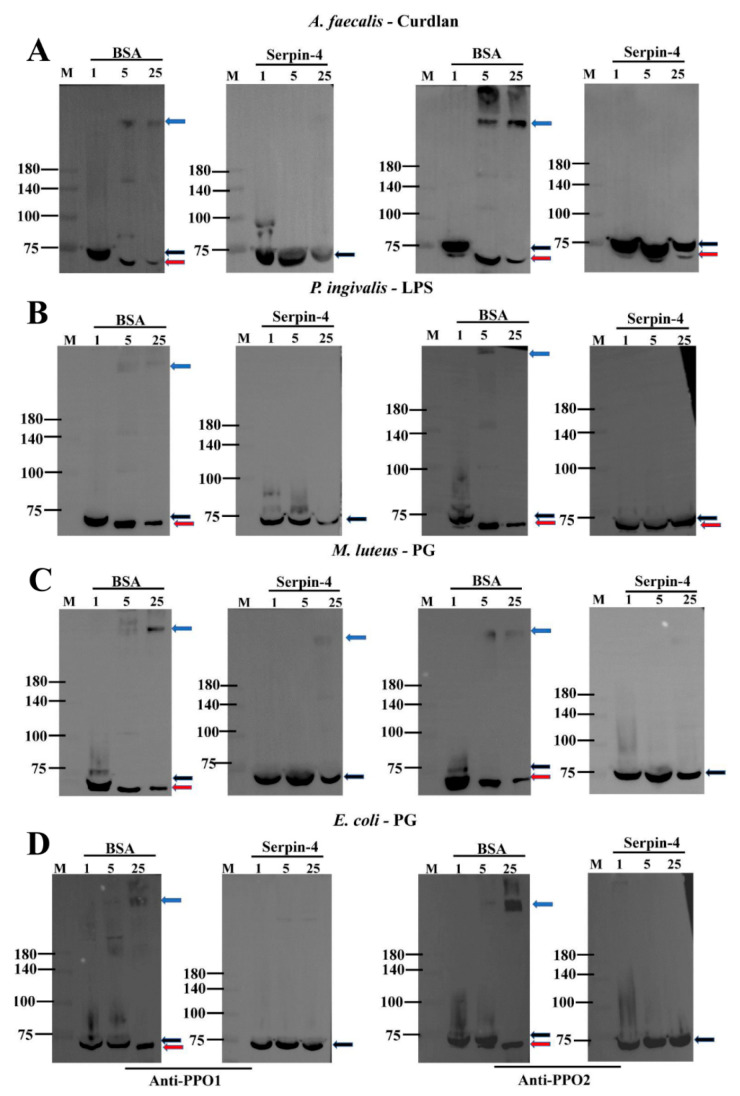
BmSerpin-4 inhibited the PPO activation of the hemolymph. Western blotting shows the clipping/activation of PPO1 (**left panel**) and PPO2 (**right panel**) and PO complex formation in the hemolymph pre-injected with BSA or BmSerpin-4 following curdlan (from *A. faecalis*) (**A**), LPS (from *P. ingivalis*) (**B**), PG (from *M. luteus*) (**C**), and PG (from *E. coli*) (**D**) activation. The blue arrows indicate the PO complexes in the plasma; the black arrows indicate PPO1 and PPO2; the red arrows indicate PO1 and PO2. M—protein molecular mass markers; 1, 5, 25—the hemolymph samples were incubated with the PAMPs for 1, 5, 25 minutes respectively.

**Figure 7 ijms-25-00313-f007:**
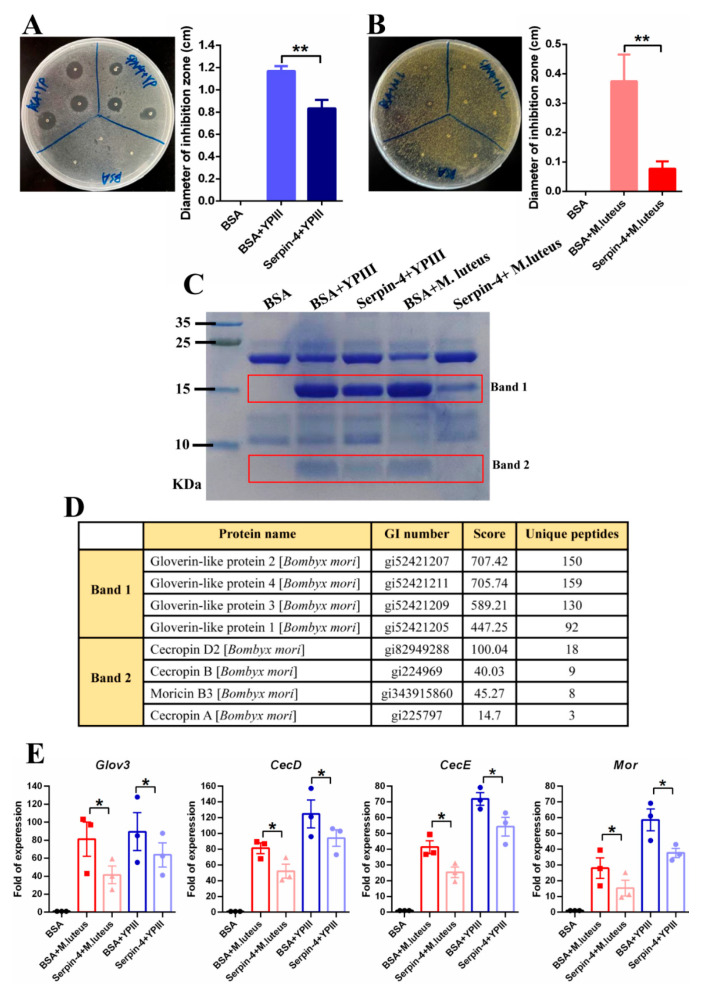
The production of AMPs was negatively regulated by BmSerpin-4. (**A**) The inhibition zone against *YP III* using the hemolymph collected from larvae pre-injected with BSA or BmSerpin-4 and then infected with *YP III*. (**B**) The inhibition zone against *M. luteus* using the hemolymph collected from the larvae pre-injected with BSA or BmSerpin-4 and then infected with *M. luteus*. For (**A**) and (**B**), the hemolymph collected from larvae injected with BSA was used as negative control. The antibacterial activity of the hemolymph was determined by the diameter of inhibition zone. (**C**) Plasma proteins were separated using tricine-SDS-PAGE and visualized with Coomassie brilliant blue staining. Note that only the lower part of the gel is shown. The boxes indicate the bands in the gel induced by bacteria. (**D**) The mass spectrometry results of the protein bands that were excised from the gel in (**C**). (**E**) The mRNA abundance of antimicrobial peptide genes *gloverin3* (*Glov 3*), *cecropin-D* (*Cec D*), *cecropin-E* (*Cec E*), and *moricin* (*Mor*) in the fat body collected from the larvae pre-injected with BSA or BmSerpin-4 and then infected with *YP III* or *M. luteus*. The fat body collected from larvae injected with BSA was used as control. Each dot in the graph represents an individual experiment. For (**A**–**C**), the values shown are the means (±SEM) of three independent experiments. The statistical differences between the compared groups are denoted with asterisks. *p*-values were determined using Student’s *t*-test. * *p* < 0.05; ** *p* < 0.01.

**Figure 8 ijms-25-00313-f008:**
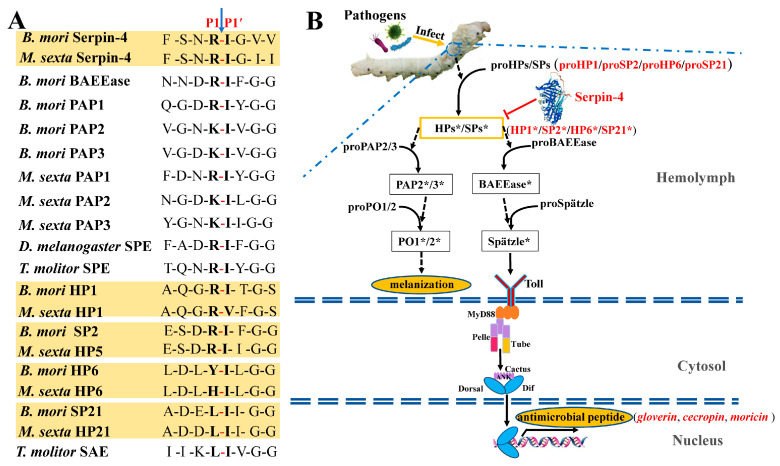
The prediction of the candidate target serine proteases for BmSerpin-4. (**A**) Comparison of activation site sequences between BmSerpin-4/MsSerpin-4 and proteases from *B. mori*, *D. melanogaster*, *M. sexta*, and *T. molitor*. PAP—prophenoloxidase-activating protease; SPE—Spätzle-processing enzyme; SAE—SPE-activating enzyme. The P1-P1′ scissile bond is indicated by an arrow. The yellow background indicates MsSerpin-4 and its corresponding identified target serine proteases, as well as BmSerpin-4 and the predicted candidate target serine proteases for BmSerpin-4. Each yellow background frame indicates a corresponding orthology between the serpins or serine proteases of *B. mori* and *M. sexta*. (**B**) A model for regulation of PPO activation and Toll pathway cascade in *B. mori* by BmSerpin-4. Arrows indicate activation of downstream components or steps. Dashed arrows indicate potentially more than one step. *: activated/clipped.

**Table 1 ijms-25-00313-t001:** Primers used for quantitative real-time PCR and gene cloning.

Primer	Sequence (5′–3′)
Quantitative real-time PCR	
Q-*Serpin-4* F	GCCATGGGTATCGAGGATTT
Q-Serpin-4 R	TTTCTGCTTTGTGTACCACTTTC
Q-*Glov3* F	GACACGAGAATGGGAGGAG
Q-*Glov3* R	AAGACCCTGGTGCCGTAA
Q-*CecD* F	CTCCCGGCAACTTCTTCA
Q-*CecD* R	CGAACCCTCTGACCCATT
Q-*CecE* F	ACTGTTCGACATCGCCTCT
Q-*CecE* R	CGAATGTTCTGACCCACC
Q-*Mor* F	CTGAAGAAGGGTGGGAAAGTTA
Q-*Mor* R	ATGAAGTCTATAGCCACGTGC
Q-*IF4A* F	TCTGGCATCATACCTTCTACAA
Q-*IF4A* R	TCTGTGTCATCTTTTCCCTGTT
Gene cloning	
*Serpin-4* F	TGCGGATCCCAAAATATTCCTAAGGCGACT
*Serpin-4* R	TGCCTCGAGTTAGTAAAGTGATGGCTCCGTA

Underlined section shows the *Bam* HI and *Xho* I restriction enzyme sites.

## Data Availability

The data presented in this study are available from the corresponding author upon reasonable request.

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
