# Peer review of "Serpin-4 Negatively Regulates Prophenoloxidase Activation and Antimicrobial Peptide Synthesis in the Silkworm, Bombyx mori"

_ijms, 2023, doi:10.3390/ijms25010313_

Round 1

Reviewer 1 Report

Comments and Suggestions for Authors

This manuscript includes a mostly well done characterization of the expression and some functional analysis of serpin-4 from Bombyx mori. The data demonstrate that this serpin is expressed at a higher level in fat body and hemocytes in response to bacterial injection, that the serpin helps to negatively regulate proPO activation and expression of some AMPs (presumably by inhibiting proteases in the activation cascades).

The work misses an opportunity to use the antiserum generated to serpin-4 to estimate its concentration in hemolymph and show that the serpin concentration in hemolymph increases after injection of bacteria. The serpin-4 concentration would also be useful in evaluation of the significance of experiments in which recombinant serpin-4 was added to plasma or injected into larvae. How much did the recombinant serpin-4 increase the total serpin-4 concentration in these experiments? The antiserum should also be used for a very important immunoprecipitation experiment to identify proteases that form complexes with serpin-4 after activation of protease cascades. I think these additional experiments should be done before accepting the paper for publication.

The manuscript needs much editing for spelling (many errors) and English writing.

Last paragraph of introduction: not inhibition of antibacterial activity. (this would mean that the serpin is inhibiting AMP). Rather decreased antibacterial activity in hemolymph compared to control.  

Fig. 1B (predicted structure) does not provide new useful information regarding the other experiments in the paper. It could be deleted. If it is kept, the signal peptide should be removed from the predicted structure.

2.2 heading (spelling error)

Fig.2 legend: give is the sample size (number of biological replicates) for each treatment. What was the post test after ANOVA to determine significance of difference between pairs of samples?

2.4   and 2.5 How much would injection of 10 ug of serpin-4 affect the hemolymph  serpin-4 concentration? The authors could use the antibody to serpin-4 to estimate by quantitative western blot (using the recombinant serpin-4 as a standard) the concentration of serpin-4 in hemolymph (in naïve and bacteria injected larvae) and determine whether injecting 10 ug would make a significant increase in concentration leading to the observed effects on melanization and antibacterial activity.

Fig. 6 does not contribute usefully to a conclusion that serpin-4 blocks PPO activation (which is shown convincingly in Fig. 5). I suggest that it be deleted, because the bands detected in western blot do not necessarily result from activation, and the experiment is not needed to support the conclusion from Fig. 5. Also, whether the antisera to PPO1 and PPO2 do not cross-react would need to be confirmed.

The speculation in the second paragraph of the discussion about candidate proteases that might be inhibited by serpin-4 should be replaced by a fairly simple experimental identification of hemolymph proteases that form high molecular weight complexes with serpin-4 (evidence for their inhibition by serpin-4). This could be done by using the serpin-4 antibody to immunoprecipitate from hemolymph plasma after incubation with Micrococcus to stimulate the protease cascade activation. The separate the immunoprecipitated proteins by SDS-PAGE and cut out the serpin-4 reactive bands at higher mass than serpin-4, and identify the proteases present by mass spectrometry. This result would greatly strengthen the manuscript.

Comments on the Quality of English Language

The paper contains many spelling errors and some unclear or incorrect English writing. These need to be corrected.

Reviewer 2 Report

Comments and Suggestions for Authors

In this manuscript, the authors examine the function of BmSerpin-4. Although this manuscript demonstrates a large amount of data, I think that it still needs a considerable revision to be acceptable (see below).

1.    The authors should provide more detailed information throughout the paper. For example, it was unclear for me why the authors focused on BmSerpin-4 and how the authors identified BmSerpin-4.

2.    I could hardly associate the arguments outlined in the text with the data shown in the figures, expecially in Result 2.4. These results needs better explanation based on figures.

Comments on the Quality of English Language

As there are many errors, the authors should carefully check the text. Belows are the examples:

Abstract Line 15 and Introduction Paragraph 4 Line 15; BmSserin-4

Introduction Paragraph 3 Line 9; in in silk gland

Reviewer 3 Report

Comments and Suggestions for Authors

In this study, the authors identified a typical serpin, BmSerpin-4, in the silkworm, and the expression of BmSerpin-4 was found to be dramatically up-regulated in the fat body and hemocytes after bacterial infections (Micrococcus luteus and Yersinia pseudotuberculosis serotype O: 3 (YP III). Pre-injection of recombinant BmSerpin-4 significantly reduced the antibacterial activity of hemolymph and the expression of AMPs (gloverin-3, cecropin-D, cecropin-E and moricin) in the fat body after bacterial infections.

The authors also observe inhibition of the melanization process, then of EO activity and PPO activation induced by BmSerpin-4.

 HP1, SP2, HP6 and SP21 were identified as candidate proteases to target BmSerpin-4 by analysis of residues adjacent to cleavage binding. Authors suggest that HP1, SP2, HP6 and SP21 might be essential in the activation of the serine protease cascade in the Toll and PPO pathways in the silkworm. In this study, the authors characterized BmSserin-4 and provided further information on the activation/control of PPO and Toll in the silkworm.

The article is very interesting and well-written, the well-described methods are clear and follow a clear experimental design, there are no critical issues, the only thing to be revised are the spaces between words and some typos, e.g. fig.2 y-axis "expereression", corrected fig. 8 M. seta.

Please, enlarge fig. 8.

Re-check the whole text and correct the spelling.

Round 2

Reviewer 2 Report

Comments and Suggestions for Authors

While most of the manuscript is well revised, I believe that for this manuscript to be acceptable, more improvements are needed with respect to comment 1.

The authors added the sentence "Bioinformatics analysis showed that BmSerpin-4 was a typical serpin protein and it might play a role in the immune system of B. mori." to the Introduction section. Was this bioinformatics analysis performed by the authors? If so, please add this information to the results section and provide more information about the analysis. If not, reference papers are required.

Author Response

Dear Professor,

We are very grateful to you for your help and suggestions on our manuscripts (ijms-2717028). Following your advices, we revised the manuscript carefully and those revised sentences were marked with blue. Our response is as below.

Comments and Suggestions for Authors

While most of the manuscript is well revised, I believe that for this manuscript to be acceptable, more improvements are needed with respect to comment 1.

The authors added the sentence "Bioinformatics analysis showed that BmSerpin-4 was a typical serpin protein and it might play a role in the immune system of B. mori." to the Introduction section. Was this bioinformatics analysis performed by the authors? If so, please add this information to the results section and provide more information about the analysis. If not, reference papers are required.

--- Tank you for your suggestion. The reference was added. Reference 21.

Kind regards

Xingtao Qie